# A tele-health primary care rehabilitation program improves self-perceived exertion in COVID-19 survivors experiencing Post-COVID fatigue and dyspnea: A quasi-experimental study

José Calvo-Paniagua[1], María José Díaz-Arribas[2,3]*, Juan Antonio Valera-Calero[4], María Isabel Gallardo-Vidal[5], César Fernández-de-las-Peñas[6], Ibai López-de-Uralde-Villanueva[2,3], Tamara del Corral[2,3], Gustavo Plaza-Manzano[2,3]

1 Gerencia Asistencial Atención Primaria de Madrid, Centro de Salud Dr. Castroviejo, Madrid, Spain, 2 Department of Radiology, Rehabilitation and Physiotherapy, Universidad Complutense de Madrid, Madrid, Spain, 3 Instituto de Investigación Sanitaria San Carlos (IdISSC), Madrid, Spain, 4 VALTRADOFI Research Group, Department of Physical Therapy, Universidad Camilo José Cela, Villanueva de la Cañada, Madrid, Spain, 5 Gerencia Asistencial Atención Primaria de Madrid, Centro de Salud Valdelasfuentes, Madrid, Spain, 6 Department of Physical Therapy, Occupational Therapy, Rehabilitation and Physical Medicine, Universidad Rey Juan Carlos, Alcorcón, Spain

* mjdiazar@med.ucm.es

## Abstract

### Background

Current evidence suggests that up to 70% of COVID-19 survivors develop post-COVID symptoms during the following months after infection. Fatigue and dyspnea seem to be the most prevalent post-COVID symptoms.

### Objective

To analyze whether a tele-rehabilitation exercise program is able to improve self-perceived physical exertion in patients with post-COVID fatigue and dyspnea.

### Methods

Sixty-eight COVID-19 survivors exhibiting post-COVID fatigue and dyspnea derived to four Primary Health Care centers located in Madrid were enrolled in this quasi-experimental study. A tele-rehabilitation program based on patient education, physical activity, airway clearing, and breathing exercise interventions was structured on eighteen sessions (3 sessions/week). Self-perceived physical exertion during daily living activities, dyspnea severity, health-related quality of life and distance walked and changes in oxygen saturation and heart rate during the 6-Minute walking test were assessed at baseline, after the program and at 1- and 3-months follow-up periods.

**Data Availability Statement:** All relevant data are within the article and its Supporting Information files.

**Funding:** Funded by the Foundation for Biosanitary Research and Innovation in Primary Care (FIIBAP) and the Regional Ministry of Health of the Community of Madrid through non-refundable grants from the credits awarded to the Community of Madrid by the Spanish Government Fund COVID-19, included in Order HAC/667/2020. Funders had no role in the study design, data collection and analysis, decision to publish nor preparation of the manuscript.

**Competing interests:** The authors have declared that no competing interests exist.

## Results

Daily living activities, dyspnea severity and quality of life improved significantly at all follow-ups (p<0.001). Additionally, a significant increase in oxygen saturation before and after the 6-Minute Walking test was found when compared with baseline (P<0.001). Heart rate adaptations at rest were found during the follow-up periods (P = 0.012). Lower perceived exertion before and after the 6-Minute Walking test were also observed, even if larger distance were walked (P<0.001).

## Conclusion

Tele-rehabilitation programs could be an effective strategy to reduce post-COVID fatigue and dyspnea in COVID-19 survivors. In addition, it could also reduce the economic burden of acute COVID-19, reaching a greater number of patients and releasing Intensive Unit Care beds for prioritized patients with a severe disease.

## Study registration

The international OSF Registry registration link is https://doi.org/10.17605/OSF.IO/T8SYB.

## Introduction

Up to date, COVID-19 caused by SARS-CoV-2 infection affected more than 450 million cases worldwide (+180 million in Europe and +145 million in the Americas) [1]. Acute manifestations heterogeneously affect the pulmonary, cardiovascular, neurologic, hematologic and gastrointestinal systems [2]. However, recent research has focused on post-acute, long-COVID or post-COVID [3–5] since the high number of COVID-19 survivors presenting post-COVID-19 sequelae represents a major health-care challenge [6]. In fact, up to the 85% of previous hospitalized COVID-19 survivors showed post-COVID-19 symptoms during the following months after the infection [7–9].

Although multiple post-COVID symptoms have been described (e.g., memory loss, brain fog, hair loss, tachycardia, pain, skin rash, gastrointestinal problems, diarrhea, anosmia, ocular problems, ageusia) [2,5–11], fatigue and dyspnea are reported as the most common symptoms developed by this population [12]. Previous research reported fatigue and dyspnea appearance 3 months after the onset in 52–58% and 24–37% of the patients respectively [12,13]. In fact, it should be noted that only 31% of the patients did not report post-COVID fatigue or dyspnea seven months after hospital discharge [12].

This startling prevalence results in important daily living impact [14]. Evidence is consistent demonstrating the association of fatigue and dyspnea with worse quality of life and greater difficulties to perform daily living activities (i.e., walking, climbing stairs or lifting) [12,14,15]. Furthermore, since no association between pre-existing comorbidities with post-COVID-19 quality of life is observed, all these functional limitations should be specifically attributed to COVID-19 [12].

In addition to the natural course of acute COVID-19, the mandatory home isolation obeyed in Spain for more than 3 months during the outbreak in March 2020 aggravated the physical conditioning of the worldwide population at different levels. In addition to a psychological impact (i.e., increase in depressive and anxiety levels) derived from the confinement [16,17], physical deterioration implies negative metabolic changes [18] and trigger peaks in type II

diabetes, both factor that could aggravate the clinical course in patients affected by COVID-19 [19].

Tele-rehabilitation programs have been widely developed during the last years (especially during the COVID-19 confinement), since is a readily accessible and feasible technology allowing long-distance communication and follow up by videoconferencing, email or texting [20]. Although currently the confinement is over, telemedicine could be considered still a feasible manner to take care of patients since this alternative clinician-patient interaction demonstrated in several disciplines to reduce the economic burden and could allow primary health care centers to reach a greater number of patients [21,22].

Since physical activity programs reported multiple gains in physical conditioning [23], implementing a tele-rehabilitation program based on exercise may reduce the rate of aggravation and hospital admissions due to fatigue and respiratory problems, improve the patients' quality of life and self-sufficiency and achieve fatigue and dyspnea benefits [24]. Therefore, our aim was to analyze whether a tele-health exercise-based program developed in Primary Health Care centers improves physical exertion in post-COVID patients. We hypothesized that the tele-health exercise program would significantly improve the self-perceived exertion and cardiovascular indicators in COVID-19 survivors with post-COVID fatigue and dyspnea.

## Methods

### Study design

This study was a prospective, multicenter, single-group, quasi-experimental study conducted between April 2020 and December 2020 in four Primary Health Care centers in Madrid (Spain) listed in the GAAP (Gerencia Asistencial de Atención Primaria). All procedures were approved by the local Ethics Committee of Hospital Universitario de la Paz (PI-4288) and conducted in accordance with the Declaration of Helsinki. In addition, the study was conducted following the Transparent Reporting of Evaluations with Nonrandomized Designs (TREND) [25] and the Enhancing the QUAlity and Transparency of health Research (EQUATOR) [26] guidelines. Since this is not a randomized clinical trial and due to the pandemic situation at the recruitment stage, we were not aware of the prospective trial registration. However, the study protocol was retrospectively registered in the International OSF Registry (registration link: https://doi.org/10.17605/OSF.IO/T8SYB). The authors confirm that all ongoing and related trials for this intervention are registered.

### Participants

Patients who had survived from SARS-CoV-2 infection and linked to one of the four GAAP centers participating in this study were screened for eligibility. Individuals aged from 25 to 65 years, who had surpassed COVID-19, with a negative PCR test at the moment of the study and reporting fatigue and dyspnea as main post-COVID symptoms from at least three months after the infection were potentially eligible. Exclusion criteria included: 1) patients with other post-COVID symptoms, e.g., gastrointestinal symptoms, anosmia, ageusia, or cognitive blurring; 2) evidence of pluri-pathology, i.e., more than two pre-existing medical comorbidities; 3, evidence of any medical co-morbidity, i.e., ischemic cardiopathy, cardiac or pulmonary insufficiency, potentially explaining fatigue or dyspnea; 4, presence of fatal medical co-morbidities e.g., cancer; 5, immunodeficient patients; 6, previous history of dementia or psychiatric disorders; 7, patients with severe functional limitations (Barthel index score > 90); or 8) patients with cognitive problems. Reading and signing the written informed consent was mandatory to be included in the study.

## Sample size calculation

Sample size estimation was calculated using the G*Power v.3.1 software for Mac OS. With the intention of detecting difference between two dependent means (change pre-post intervention), an a priori analysis was conducted by running a paired t-test as a statistical test. The input parameters were set for bilateral contrast (two-tailed), α = 0.05, β = 0.05 (95% power). According to the criteria established by Cohen [27], an effect size of moderate magnitude is required to detect clinically relevant differences. Thus, a moderate effect size (d = 0.5) was chosen. These data lead to a minimum sample size of 54 participants. Due to the longitudinal nature of this study, an additional 10% sample size was included. Therefore, a sample size of 60 participants was proposed as appropriate.

## Intervention

The exercise-based rehabilitation program was performed in a tele-health modality by video-conference using Zoom. This program was divided in 18 sessions of 40 minutes of duration each one, three times a week (alternating days). Therefore, the total duration of the program was planned to be up to 7 weeks.

The program consisted of 1) sessions of sanitary education (introducing the patients to concepts of anatomy, physiology, primary prevention aspects related with COVID-19 including hygienic habits, use of facemasks and social distance, smoking cessation, weight control, nutrition, benefits of regular physical activity) and posture ergonomics; 2) respiratory control, diaphragmatic respiration education, volume-targeted ventilation, secretion clearance and exercises targeting the respiratory muscles; 3) specific physical conditioning targeting the spine, respiratory, core and lower and upper extremity muscles performing aerobic exercise trainings, active mobilizations and motor control exercises. Details of the complete program are available in **Table 1**.

## Outcomes

The following outcomes were evaluated at baseline, at the end of the rehabilitation program, and one and three months after (follow-up periods).

The perceived physical exertion during their daily living activities was assessed with the Modified Borg Dyspnea Scale (MBDS), a valid and reliable method for assessing dyspnea in patients with respiratory conditions [28]. Participants were asked to point in a vertical scale ranging from 0 to 10 (in which numbers are anchored with corresponding verbal expressions of progressively increasing intensity from no breathlessness at all to maximal breathlessness) their mean self-perceived exertion during daily living tasks [28].

Dyspnea severity was classified using the modified Medical Research Council (mMRC) scale as is one of the most widely used and validated scale to assess dyspnea in daily living in chronic respiratory diseases [29]. Participants were classified as Grade 0 ("breathless with strenuous exercise"), Grade 1 ("short of breath while hurrying on level ground or walking up a hill"), Grade 2 ("walk slower than people of the same age on level ground and experience breathlessness or the need to stop if walking on level ground at their own pace"), Grade 3 ("stop to breathe after walking few minutes on level ground") or Grade 4 ("too breathless to leave the house or during non-breathless activities like dressing or undressing") [29].

Health-related quality of life was assessed by using the St George's Respiratory Questionnaire (SGRQ). The SGRQ was developed for patients with asthma and chronic obstructive pulmonary disease and consists of 50-items divided into three domains for measuring symptoms, activity limitations and the psychosocial impact [30]. Final scores range from 0 (best health status) to 100 (poorest health status) [30].

**Table 1. Tele-rehabilitation program chronogram.**

| | |
|---|---|
| Session 1 | Theoretical session:<br>• Basic principles of anatomy and physiology<br>• COVID-19 preventive measures (safe distance, use of facemasks, room ventilation and hand hygienization)<br>• Sanitary education<br>• Postural ergonomics. |
| Session 2–5 | Breathing exercises:<br>• Diaphragmatic breathing, costal breathing, pursed-lips breathing and airways cleaning |
| Session 6–8 | Breathing exercises<br>Phyisical conditioning with increasing intensity:<br>• Cervical, dorsal and lumbar spine active mobilizations<br>• Lower and upper limb active mobilizations<br>• Core training with motor control exercises |
| Session 9 | Breathing exercises<br>Physical conditioning with increasing intensity<br>Body balance training:<br>• Dynamic sitting control exercises<br>• Deambulation exercises |
| Session 10–11 | Breathing exercises<br>Physical conditioning with increasing intensity<br>Functional exercises:<br>• Plyometric exercises<br>Occupational therapy exercises for daily living activities |
| Session 12–18 | Breathing exercises<br>Physical conditioning with increasing intensity<br>Functional exercises<br>Occupational therapy exercises<br>Aerobic training:<br>• Walking at tolerable speed |

Finally, the 6-Minute Walking Test (6MWT) was performed to evaluate whether this physical demand change their heart rate, $O_2$ saturation, perceived physical exertion (using again the MDBS) and distance walked [31]. Participants were instructed that the objective of this test was to walk the larger distance as possible during 6 minutes in a flat, long and covered corridor (30m long approximately) marked each meter for facilitating the distance calculations [31].

## Statistical analysis

All statistical analyses were conducted with in the SPSS Statistics software v.25 (IBM Corporation, Armonk, NY, USA), setting a significance level of p<0.05 for all tests. Data distribution was verified using the Saphiro-Wilk test and histograms. Levene tests were used for variance homogeneity. Descriptive analyses were performed to characterize the sample. Central tendency and dispersion data were reported as mean and standard deviation for normal-distributed variables, or as median and interquartile range for non-normal-distributed variables, respectively. Within-group differences were assessed using a lineal mixed model for controlling cofounding effects due to the lack of randomization, including the dependent variables and the time-point (baseline, immediately after, and 1 and 3 months after) as the fix factor. Due to the use of multiple comparisons in the 6 outcomes assessed, the Bonferroni correction was applied Accordingly, P values were assumed to be significant at <0.0083 (0.05/6). Finally, the effect size was estimated using the $\eta_p^2$ if significant. An effect size of 0.01 was considered small, 0.06 medium and 0.14 large.

## Results

From 78 volunteers screened for eligibility criteria, a total of 71 were initially included. Three participants withdrawn from the study during the follow-up for unknown reasons (one during

the follow-up at 1 month and 2 during the follow-up at 3 months), therefore, 68 were finally analyzed (**Fig 1**. CONSORT Flow Diagram). No adverse events were reported. The mean duration of the post-COVID symptoms was 4.7 (SD 0.5) months after hospital discharge. **Table 2** provides sociodemographic features, clinical, and hospitalization data of the total sample.

**Table 3** summarizes the evolution of self-perceived physical exertion during daily living activities (MBDS), dyspnea severity (mMRC), and respiratory-related quality of life (SGRQ). The ANOVA interaction effect analysis revealed significant improvements with large effect sizes in all outcomes at all follow-up periods (P<0.001).

**Table 4** describes the changes in heart rate and oxygen saturation before and after the 6-MWT at baseline, post- and during the follow-up. A significant reduction in the pre-test heart rate was found (P = 0.001 $\eta^2_p$ = 0.062). Although this change was not significant after the intervention (P = 0.580) or 1 moth (P = 0.013), significant heart rate adaptations were seen during the follow-ups at 3 (P = 0.001) months. Similarly, although changes in heart rate were significantly different with large effect size (P = 0.001; $\eta^2_p$ = 0.059), and no post-intervention differences were found, the follow up at 1 (P = 0.004) and 3 (P = 0.002) months showed greater changes in heart rate compared with baseline (**Table 4**).

Regarding $O_2$ saturation, significant changes were found prior (P<0.001; $\eta^2_p$ = 0.260) and after (P<0.001; $\eta^2_p$ = 0.170) the 6-MWT. Results showed a significant increase after the intervention (pre-test P<0.001; post-test p = 0.001) and during the follow-up at 1 (pre-test p<0.001; post-test p<0.001) and 3 (pre-test p<0.001; post-test p<0.001) months. However, the $O_2$ Saturation change during the test did not show significant differences (P = 0.248, **Table 4**),

**Table 5** shows the evolution in distance walked and perceived exertion change during the 6-MWT. Participants decreased significantly their perceived exertion before starting the test (P<0.001; $\eta^2_p$ = 0.835) and after finishing the test (P<0.001; $\eta^2_p$ = 0.337). The Borg scale difference also revealed a significant improvement (P<0.001; $\eta^2_p$ = 0.065). Pre-test scores, post-test scores and changes showed a significant improvement after the intervention (all, P<0.001). Despite the follow-ups demonstrated significant reductions in pre-test and post-test scores (P<0.001), the change was comparable with baseline data (1-month, P = 0.126; 3-months, P = 1.000). Finally, the distance walked during the test was significantly greater (P<0.001; $\eta^2_p$ = 0.065). Compared with baseline, the distance walked was significantly larger after finishing the intervention, after 1 month and after 3 months (all, P<0.001).

## Discussion

This study analyzed whether a tele-rehabilitation exercise program was able to improve physical exertion in COVID-19 survivors exhibiting post-COVID fatigue and dyspnea. This program was based on a previous proposal provided by Wang et al. [31] performing patient education sessions, physical activity, airway clearing and breathing exercises.

In general, we found significant improvements in self-perceived physical exertion during daily living activities, dyspnea, quality of life after finishing the rehabilitation program, changes which were maintained up to 3 months after. Although there were no heart rate adaptations after finishing the rehabilitation program, basal heart rate showed a significant decrease during the follow-up and the heart rate change during the 6-MWT increased. Regarding the $O_2$ Saturation, basal and post-test levels increased significantly. Finally, the self-perceived physical exertion before and after the 6-MWT decreased after the intervention and was maintained up to 3 months while larger distances were walked.

These results reinforce the recommendations for implementing tele-rehabilitation programs [21–24] since this is a feasible way to manage and follow a higher number of patients

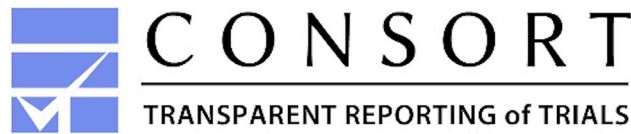

## CONSORT 2010 Flow Diagram

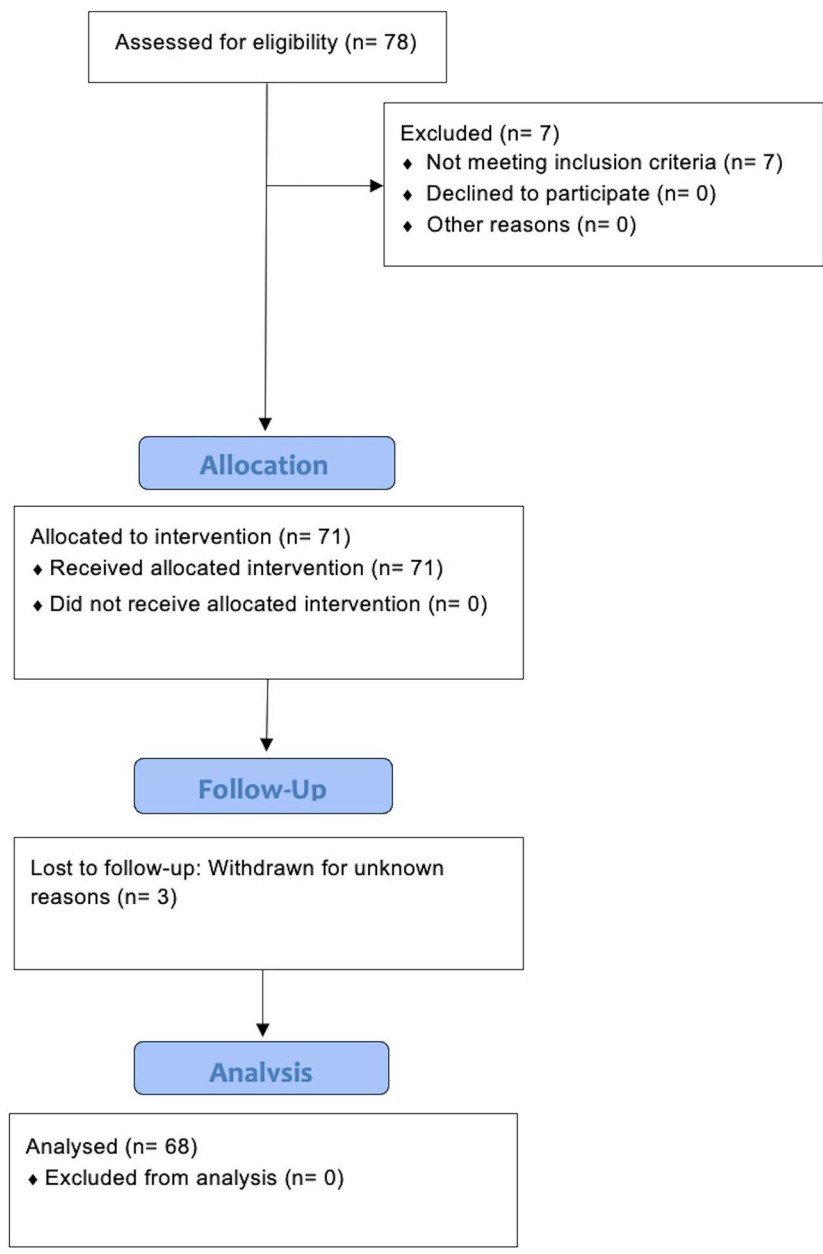

**Fig 1. CONSORT 2010 flow diagram.**

**Table 2. Sociodemographic and clinical data of the sample at baseline (n = 68).**

| Sociodemographic characteristics | |
|---|---|
| Age, mean ± SD, years | 48.5 ± 9.7 |
| Gender, male/female, n (%) | 26 (38.2) / 42 (61.8) |
| Weight, mean ± SD, kg | 79.7 ± 18.1 |
| Smoking, yes/no/ex, n (%) | 8 (11.8) / 41 (60.3) / 19 (27.9) |
| *Pre-existing comorbidities* | |
| AHT, yes/no, n (%) | 1 (1.5) / 67 (98.5) |
| Diabetes, yes/no, n (%) | 6 (8.8) / 62 (91.2) |
| Obesity, yes/no, n (%) | 34 (50.0) / 34 (50.0) |
| COPD, yes/no, n (%) | 0 (0.0) / 68 (100.0) |
| Asthma, yes/no, n (%) | 8 (11.8) / 60 (88.2) |
| *Health care at COVID onset* | |
| Hospitalization yes/no, n (%) | 30 (44.1) / 38 (55.9) |
| Duration, mean ± SD, days | 7.7 ± 5.6 |
| ICU admission, yes/no, n (%) | 3 (4.4) / 65 (95.6) |
| Duration, mean ± SD, days | 9.0 ± 4.6 |

AHT: Arterial Hypertension; COPD: Chronic Obstructive Pulmonary Disease; ICU: Intensive Care Unit.

than traditional face-to-face visits and could significantly decrease the economic burden of management of post-COVID patients in primary health care centers. However, this modality of treatment cannot be used for all post-COVID patients, since individuals showing moderate to severe symptoms (i.e., respiratory distress with respiratory rate >30 times/minutes, $O_2$ saturation <93% or $PaO_2/FiO_2 < 300mmHg$) require hospitalization and monitoring [32], this is a feasible alternative for patients with mild symptomatology to prioritize the limited number of beds in the intensive care unit (ICU) for patients with moderate to severe acute COVID-19 symptoms. Therefore, since it has been reported that almost 80% of COVID-19 patients do not require management at ICU during the acute phase of the infection [33], further research should focus on efficient rehabilitation and exercise strategies to offload ICUs to reduce the

**Table 3. Perceived physical exertion during daily living activities, dyspnea severity and health-related quality of life.**

| Variable | Modified Borg Scale (0–10) | mMRC Scale (0–5) | SGRQ (0–100) |
|---|---|---|---|
| **Baseline** | 7.3 ± 1.4 | 2.57 ± 0.65 | 55.6 ± 15.2 |
| **Post-Intervention** | 0.9 ± 1.1 | 0.17 ± 0.38 | 11.8 ± 5.3 |
| **Follow-up: 1 Month** | 0.2 ± 0.6 | 0.02 ± 0.17 | 7.6 ± 3.4 |
| **Follow-up: 3 Months** | 0.1 ± 0.4 | 0.04 ± 0.26 | 7.5 ± 5.2 |
| *ANOVA interaction effect* | | | |
| | F = 862.731 $\eta^2_p = 0.906$ p<0.001 | F = 634.942 $\eta^2_p = 0.876$ p<0.001 | F = 497.053 $\eta^2_p = 0.847$ p<0.001 |
| *Within-Group Differences*** | | | |
| Post-intervention | 6.3 (5.9;6.8) * | 2.40 (2.21;2.59) * | 43.7 (39.8;47.7) * |
| 1 Month | 7.0 (6.6;7.5) * | 2.55 (2.36;2.73) * | 47.9 (44.0;51.8) * |
| 3 Months | 7.2 (6.7;7.6) * | 2.53 (2.34;2.72) * | 48.0 (44.1;52.0) * |

mMRC: Modified British Medical Research Council; SGRQ: Saint George's Respiratory Questionnaire.

* Significant differences (P <0.001).

** Compared with baseline scores.

**Table 4. Heart rate and oxygen saturation changes during the 6-Minute Test.**

| | Heart Rate (bpm) | | | O$_2$ Saturation (%) | | |
|---|---|---|---|---|---|---|
| | Pre-Test | Post-Test | Change | Pre-Test | Post-Test | Change |
| **Baseline** | 76.1 ± 12.4 | 122.7 ± 14.9 | 46.6 ± 16.2 | 96.4 ± 1.5 | 96.2 ± 2.0 | -0.1 ± 2.0 |
| **Post-Intervention** | 73.5 ± 10.9 | 125.1 ± 16.5 | 50.8 ± 16.3 | 97.7 ± 1.3 | 97.2 ± 1.4 | -0.4 ± 1.6 |
| **Follow-up: 1 Month** | 70.8 ± 8.8 | 125.5 ± 9.7 | 54.7 ± 10.6 | 98.2 ± 1.1 | 97.6 ± 1.1 | -0.6 ± 1.3 |
| **Follow-up: 3 Months** | 69.5 ± 7.3 | 124.5 ± 6.5 | 55.1 ± 8.8 | 98.2 ± 0.9 | 98.1 ± 1.6 | -0.1 ± 1.0 |
| | *ANOVA interaction effect* | | | | | |
| | F = 5.937 | F = 0.656 | F = 5.636 | F = 31.435 | F = 18.315 | F = 1.383 |
| | $\eta^2_p = 0.062$ | $\eta^2_p = 0.007$ | $\eta^2_p = 0.059$ | $\eta^2_p = 0.260$ | $\eta^2_p = 0.170$ | $\eta^2_p = 0.015$ |
| | p = 0.001 | p = 0.580 | p = 0.001 | p<0.000 | p<0.000 | p = 0.248 |
| | *Within-Group Differences** | | | | | |
| Post-intervention | -2.6 (-7.2;2.0) p = 0.789 | N/A | 5.0 (-1.2;11.2) P = 0.201 | 1.3 (0.7;1.8) P<0.001 | 0.9 (0.3;1.6) P = 0.001 | N/A |
| 1 Month | -5.3 (-9.9;-0.8) p = 0.012 | | 8.1 (1.9;14.3) P = 0.004 | 1.7 (1.2;2.3) P<0.001 | 1.3 (0.6;2.0) P<0.001 | |
| 3 Months | -6.7 (-11.2;-2.1) P = 0.001 | | 8.5 (2.3;14.8) P = 0.002 | 1.8 (1.2;1.3) P<0.001 | 1.8 (1.1;2.4) P<0.001 | |

* Compared with Pre-test scores.

burden and the impact of COVID-19 in patients' quality of life and function even if limited resources are available.

Finally, some limitations in this study should be acknowledged. First, the quasi-experimental design of this study has no control group. It should be considered that those patients recruited for this study were recruited during the first worldwide outbreak, hence, the inclusion of a control group without intervention was not considered ethic. However, this must be recognized as a limitation since there is a potential bias that the improvement observed in these patients may not be totally attributable to the exercise program. Current knowledge on

**Table 5. Perceived exertion and distance walked during the 6-Minute Test.**

| | Modified Borg Scale (0–10) | | | Distance (m) |
|---|---|---|---|---|
| | Pre-Test | Post-Test | Change | |
| **Baseline** | 6.8 ± 2.0 | 8.6 ± 1.5 | 1.8 ± 1.2 | 560.1 ± 98.9 |
| **Post-Intervention** | 1.0 ± 1.2 | 5.1 ± 6.5 | 4.0 ± 6.5 | 638.1 ± 95.5 |
| **Follow-up: 1 Month** | 0.1 ± 0.4 | 3.3 ± 1.8 | 3.1 ± 1.8 | 672.2 ± 72.5 |
| **Follow-up: 3 Months** | 0.1 ± 0.6 | 2.1 ± 1.4 | 2.0 ± 1.1 | 699.7 ± 57.2 |
| | *ANOVA interaction effect* | | | |
| | F = 452.524 | F = 45.508 | F = 6.213 | F = 36.469 |
| | $\eta^2_p = 0.835$ | $\eta^2_p = 0.337$ | $\eta^2_p = 0.065$ | $\eta^2_p = 0.289$ |
| | P< 0.001 | P< 0.001 | P< 0.001 | p<0.001 |
| | *Within-Group Differences** | | | |
| Post-intervention | -5.7 (-6.3;-5.2) P< 0.001 | -3.5 (-5.1;-1.9) P< 0.001 | 2.2 (0.6;3.8) P = 0.001 | 78.0 (40.4;115.7) P<0.001 |
| 1 Month | -6.7 (-7.3;-6.1) P< 0.001 | -5.3 (-6.9;-3.7) P< 0.001 | 1.3 (-0.2;3.0) P = 0.126 | 112.0 (74.4;149.7) P<0.001 |
| 3 Months | -6.7 (-7.3;-6.2) P< 0.001 | -6.5 (-8.1;-4.9) P< 0.001 | 0.2 (-1.4;1.8) P = 1.000 | 139.6 (101.9;177.2) P<0.001 |

* Compared with Pre-test scores.

post-COVID supports that individuals can exhibit long-lasting fatigue and dyspnea up to one year after the infection [7–9], which would support that the effects could be attributable to the program. Secondly, since this is a tele-rehabilitation program, it was not possible to control some essential aspects of the intervention e.g., the intensity of aerobic exercise. Finally, the sample analyzed in this study could be considered as small. Further research including a control group, administering controlled interventions with larger sample sizes and follow-up periods are needed to further corroborate current results.

## Conclusion

The application of a tele-rehabilitative primary care exercise program based on patient education, physical activity, airway clearing and breathing exercise was able to improve self-perceived physical exertion in COVID-19 survivors reporting post-COVID fatigue and dyspnea. Further experimental clinical trials including a comparative group, specific control on exercise dosage and administration and larger samples and follow-up periods are needed to corroborate these findings.

## Supporting information

**S1 Checklist. TREND statement checklist.**
(PDF)

**S1 File. Trial study protocol english.**
(DOCX)

**S2 File. Trial study protocol original.**
(DOCX)

## Acknowledgments

Disclosures

### Registration

The international OSF Registry registration link is https://doi.org/10.17605/OSF.IO/T8SYB.

## Author Contributions

**Conceptualization:** José Calvo-Paniagua, María José Díaz-Arribas, Gustavo Plaza-Manzano.

**Data curation:** Juan Antonio Valera-Calero.

**Funding acquisition:** José Calvo-Paniagua.

**Investigation:** María José Díaz-Arribas, Juan Antonio Valera-Calero, María Isabel Gallardo-Vidal, Ibai López-de-Uralde-Villanueva, Tamara del Corral, Gustavo Plaza-Manzano.

**Methodology:** José Calvo-Paniagua, María José Díaz-Arribas, Juan Antonio Valera-Calero, María Isabel Gallardo-Vidal, César Fernández-de-las-Peñas, Ibai López-de-Uralde-Villanueva, Tamara del Corral, Gustavo Plaza-Manzano.

**Project administration:** José Calvo-Paniagua.

**Resources:** José Calvo-Paniagua, María Isabel Gallardo-Vidal.

**Supervision:** María José Díaz-Arribas, Gustavo Plaza-Manzano.

**Writing – original draft:** María José Díaz-Arribas, Juan Antonio Valera-Calero, César Fernández-de-las-Peñas.

**Writing – review & editing:** José Calvo-Paniagua, Juan Antonio Valera-Calero, María Isabel Gallardo-Vidal, César Fernández-de-las-Peñas, Ibai López-de-Uralde-Villanueva, Tamara del Corral, Gustavo Plaza-Manzano.

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
