## [Decision Letter · Decision Letter 0]

12 May 2022

PONE-D-22-08829A Tele-Presential Primary Care Rehabilitation Program Improves Self-Perceived Exertion in COVID-19 Survivors experiencing Post-COVID Fatigue and Dyspnoea: A Quasi-Experimental StudyPLOS ONE

Dear Dr. Díaz-Arribas,

Thank you for submitting your manuscript to PLOS ONE. After careful consideration, we feel that it has merit but does not fully meet PLOS ONE’s publication criteria as it currently stands. Therefore, we invite you to submit a revised version of the manuscript that addresses the points raised during the review process.

Both reviewers highlighted major problems in the statistical methodology and in the lack of a control group. here are proposed ways to overcome this last major problem as suggested by the reviewer. 

We look forward to receiving your revised manuscript.

Kind regards,

Andrea Martinuzzi

Academic Editor

PLOS ONE

Journal Requirements:

2. Thank you for submitting your clinical trial to PLOS ONE and for providing the name of the registry and the registration number. The information in the registry entry suggests that your trial was registered after patient recruitment began. PLOS ONE strongly encourages authors to register all trials before recruiting the first participant in a study.

1) your reasons for your delay in registering this study (after enrolment of participants started);

2) confirmation that all related trials are registered by stating: “The authors confirm that all ongoing and related trials for this drug/intervention are registered

6. Please upload a new copy of Figure 1 as the detail is not clear. Please follow the link for more information: https://blogs.plos.org/plos/2019/06/looking-good-tips-for-creating-your-plos-figures-graphics/" https://blogs.plos.org/plos/2019/06/looking-good-tips-for-creating-your-plos-figures-graphics/

Reviewers' comments:

Reviewer's Responses to Questions

**Comments to the Author**

1. Is the manuscript technically sound, and do the data support the conclusions?

Reviewer #1: Partly

Reviewer #2: Partly

2. Has the statistical analysis been performed appropriately and rigorously? 

Reviewer #1: Yes

Reviewer #2: No

3. Have the authors made all data underlying the findings in their manuscript fully available?

Reviewer #1: Yes

Reviewer #2: Yes

4. Is the manuscript presented in an intelligible fashion and written in standard English?

Reviewer #1: Yes

Reviewer #2: Yes

5. Review Comments to the Author

Reviewer #1: There is no control group which is a major weakness. This could be overcome by looking at the same intervention in nonCOVID patients, or those who refused the interventions, or historic controls from other respiratory conditions. The paper also does not reflect the literature demonstrating that the majority of symptoms resolve in 3 months which is why the WHO identify post-Covid conditions in those whose symptoms persist more than 90 days. Without any sense of whether these symptoms would have resolved anyway, it is hard to make any assessment of the statistical improvements documented by the interventions.

Minor suggestions - spell dyspnea consistently throughout

Change the title word "tele-Presential" as this has no apparent meaning, maybe tele-health etc

Reviewer #2: The sample size calculation is not correct. This is an one-arm longitudinal study and the primary analysis focused on pre-post change. One-sample t-test is not correct. Further, the effect size of 0.5 needs to be justified.

There are 6 outcomes analyzed. May divide alpha by 6 for multiple test adjustment.

Since there is no randomization, confounding effects may need to be controlled. A linear mixed model can be used.

6. PLOS authors have the option to publish the peer review history of their article (what does this mean?). If published, this will include your full peer review and any attached files.

Reviewer #1: No

Reviewer #2: No

---

## [Author Response · Author response to Decision Letter 0]

3 Jun 2022

Response Letter manuscript ID 8464699

A Tele-Health Primary Care Rehabilitation Program Improves Self-Perceived Exertion in COVID-19 Survivors experiencing Post-COVID Fatigue and Dyspnoea: A Quasi-Experimental Study

We would like to thank the reviewers for their comments, which we believe have clarified many aspects of the manuscript. We have edited the text according to the suggestions from the reviewers. We have highlighted all changes in yellow throughout the manuscript. A point-by-point response is presented below.

Reviewer 1

There is no control group which is a major weakness. 

Response: We agree with the reviewer and this weakness is included in the limitation section as the first one (lines 273-277)

This could be overcome by looking at the same intervention in non-COVID patients, or those who refused the interventions, or historic controls from other respiratory conditions. 

Response: We believe that comparing the same intervention in non-COVID patients would have no sense since these individuals would not exhibit post-COVID symptoms. The same rationale could be applied to other respiratory conditions since COVID-19 and post-COVID are different from previous respiratory conditions. Finally, we believe that, at the moment of the first outbreak when this study was conducted, a control group consisting of individuals with post-COVID symptoms but not receiving treatment would be not ethically appropriated. Otherwise, the lack of a control group is included in the limitation section as appropriate. 

The paper also does not reflect the literature demonstrating that the majority of symptoms resolve in 3 months which is why the WHO identify post-Covid conditions in those whose symptoms persist more than 90 days. Without any sense of whether these symptoms would have resolved anyway, it is hard to make any assessment of the statistical improvements documented by the interventions.

Response: We respectfully disagree with this comment from the reviewer. The literature does not demonstrate that the majority of symptoms resolve in 3 months after the infection. On the contrary, the literature clearly supports the opposite, that up to 60% of patients exhibit post-COVID symptoms six months after the infection (J Infect Dis. 2022 Apr 16:jiac136) and that almost 30% still continue with symptoms up to one year after the infection (Clin Microbiol Infect. 2022; 28: 657-66; Pathogens. 2022; 11: 269).

As the reviewer has pointed out, the WHO definition of post-COVID includes that symptoms should be present for at lest three months after the infection. Obviously, all of our patients satisfied this criterion; otherwise they would be not included in this study. As the reviewer can see, the study was conducted between April and December 2020, since all participants included in the study were those attending their GP for their post-COVID symptoms longer than 3 months after hospitalization. We recognize that this topic was not clearly stated in the previous version of the text and we have clarified now as follows (line 128) and also in the results (lines 212-214):

Line 128: “Individuals aged from 25 to 65 years, who had surpassed COVID-19, with a negative PCR test at the moment of the study and reporting fatigue and dyspnea as main post-COVID symptoms from at least three months after the infection were potentially eligible”

Lines 212-214: “The mean duration of the post-COVID symptoms was 4.7 (SD 0.5) months after hospital discharge.”

Minor suggestions - spell dyspnea consistently throughout

Response: We have modified all terms to “dyspnea” to be consistent.

Change the title word "tele-Presential" as this has no apparent meaning, maybe tele-health etc.

Response: We have modified as suggesting in the title and throughout text. 

Reviewer 2

The sample size calculation is not correct. This is a one-arm longitudinal study and the primary analysis focused on pre-post change. One-sample t-test is not correct. Further, the effect size of 0.5 needs to be justified.

Response: Thank you for this good comment. As this is stated, our primary objective was to analyze pre-post change, so the test selected to determine the sample size calculation was chosen based on this purpose. In this regard, we chose a related samples t-student test. In accordance with your comment, we have modified the sentence of the selected statistical test to clarify for the reader the selected statistical test and the reasons for its choice.

On the other hand, regarding the rationale for the effect size, we decided to select a modern effect size since, according to Cohen's criteria, it is the magnitude necessary to be able to detect statistically significant differences. This theoretical criterion was chosen to determine the effect size, since we lacked previous evidence on which to base our selection of effect size due to the fact that COVID is a “recent” condition.

See lines 139-144: “With the intention of detecting difference between two dependent means (change pre-post intervention), an a priori analysis was conducted by running a paired t-test as a statistical test. The input parameters were set for bilateral contrast (two-tailed), �=0.05, �=0.05 (95% power). According to the criteria established by Cohen [27], an effect size of moderate magnitude is required to detect clinically relevant differences. Thus, a moderate effect size (d=0.5) was chosen.”

There are 6 outcomes analyzed. May divide alpha by 6 for multiple test adjustment.

Response: We agree with the reviewer and we have now modified this as follows (lines 201-203):

“Due to the use of multiple comparisons in the 6 outcomes assessed, the Bonferroni correction was applied Accordingly, P values were assumed to be significant at <0.0083 (0.05/6).”

Since there is no randomization, confounding effects may need to be controlled. A linear mixed model can be used.

Response: We revised the statistical analyses in accordance with your recommendations. 

We thank the reviewer and we hope that the current version of the paper can be accepted in PLOS ONE

Sincerely yours, 

The authors

---

## [Decision Letter · Decision Letter 1]

8 Jul 2022

A Tele-Health Primary Care Rehabilitation Program Improves Self-Perceived Exertion in COVID-19 Survivors experiencing Post-COVID Fatigue and Dyspnoea: A Quasi-Experimental Study

PONE-D-22-08829R1

Dear Dr. Díaz-Arribas,

We’re pleased to inform you that your manuscript has been judged scientifically suitable for publication and will be formally accepted for publication once it meets all outstanding technical requirements.

Kind regards,

Andrea Martinuzzi

Academic Editor

PLOS ONE

Additional Editor Comments (optional):

Reviewers' comments:

Reviewer's Responses to Questions

**Comments to the Author**

1. If the authors have adequately addressed your comments raised in a previous round of review and you feel that this manuscript is now acceptable for publication, you may indicate that here to bypass the “Comments to the Author” section, enter your conflict of interest statement in the “Confidential to Editor” section, and submit your "Accept" recommendation.

Reviewer #1: All comments have been addressed

Reviewer #2: All comments have been addressed

2. Is the manuscript technically sound, and do the data support the conclusions?

Reviewer #1: Yes

Reviewer #2: (No Response)

3. Has the statistical analysis been performed appropriately and rigorously? 

Reviewer #1: I Don't Know

Reviewer #2: (No Response)

4. Have the authors made all data underlying the findings in their manuscript fully available?

Reviewer #1: Yes

Reviewer #2: (No Response)

5. Is the manuscript presented in an intelligible fashion and written in standard English?

Reviewer #1: Yes

Reviewer #2: (No Response)

6. Review Comments to the Author

Reviewer #1: acceptable revisions on main issues

My comment in the initial review regarding 3 months was not expressed clearly but this revision addresses the issues.

Reviewer #2: (No Response)

7. PLOS authors have the option to publish the peer review history of their article (what does this mean?). If published, this will include your full peer review and any attached files.

Reviewer #1: No

Reviewer #2: No

---

## [Editor Report · Acceptance letter]

28 Jul 2022

PONE-D-22-08829R1 

A Tele-Health Primary Care Rehabilitation Program Improves Self-Perceived Exertion in COVID-19 Survivors experiencing Post-COVID Fatigue and Dyspnea: A Quasi-Experimental Study 

Dear Dr. Díaz-Arribas:

I'm pleased to inform you that your manuscript has been deemed suitable for publication in PLOS ONE. Congratulations! Your manuscript is now with our production department. 

Kind regards, 

on behalf of

Dr. Andrea Martinuzzi 

Academic Editor

PLOS ONE